# Opinion: Stratospheric Ozone – Depletion, Recovery and New Challenges

Martyn P. Chipperfield[1,2] and Slimane Bekki[3]

[1]School of Earth and Environment, University of Leeds, LS2 9JT, UK
[2]National Centre for Earth Observation, University of Leeds, Leeds LS2 9JT, UK
[3]LATMOS/IPSL Sorbonne Université, UVSQ, CNRS, 75005 Paris, France

*Correspondence to*: Martyn P. Chipperfield (M.Chipperfield@leeds.ac.uk)

**Abstract.** We summarise current important and well-established open issues related to the depletion of stratospheric ozone and discuss some newly emerging challenges. The ozone layer is recovering from the effects of halogenated source gases due to the continued success of the Montreal Protocol despite recent renewed production of controlled substances and the impact of uncontrolled very short-lived substances. The increasing atmospheric concentrations of greenhouse gases such as carbon dioxide, methane ($CH_4$) and nitrous oxide ($N_2O$) have large potentials to perturb stratospheric ozone but their future evolutions, and hence impacts, are uncertain. Ozone depletion through injection of smoke particles has been observed following recent Australian wildfires. Further perturbations to the ozone layer are currently occurring through the unexpected injection of massive amounts of water vapour from the Hunga Tonga-Hunga Ha`apai volcano in 2022. Open research questions emphasise the critical need to maintain, if not expand, the observational network and to address the impending 'satellite data gap' in global, height-resolved observations of stratospheric trace gases and aerosols. We will, in effect, be largely blind to the stratospheric effects of similar wildfire and volcanic events in the near future. Complex Earth System Models (ESMs) being developed for climate projections have the stratosphere as an important component. However, the huge computational requirement of these models must not result in an oversimplification of the many processes affecting the ozone layer. Regardless, a hierarchy of simpler process models will continue to be important for testing our evolving understanding of the ozone layer and for providing policy-relevant information.

## 1 Introduction

Depletion of the stratospheric ozone layer has been a major environmental issue of the past few decades, especially since the discovery of the Antarctic ozone hole in 1985 (Farman et al., 1985). The observed depletion at middle and high latitudes has been caused by increasing abundances of chlorine and bromine species, which are derived from long-lived surface-emitted halogenated gases, so-called ozone-depleting substances (ODSs) (see e.g. Solomon, 1999). A primary reason for concern is that the ozone layer prevents harmful, biologically damaging ultraviolet (UV) radiation (wavelengths below about 300 nm) from reaching the surface. UV radiation can, among other impacts, cause skin cancer in humans and can be damaging to plants (Barnes et al., 2019). Ozone not only absorbs UV radiation, heating up the stratosphere, but also interacts with terrestrial infrared (IR) radiation (e.g. Riese et al., 2012). As such, it plays a key role in determining the temperature structure of the atmosphere. Hence, changes in the ozone layer can also affect surface climate, and moreover the long-lived ODSs, such as chlorofluorocarbons (CFCs), themselves are also potent greenhouse gases (Velders et al., 2007).

The Montreal Protocol on Substances that Deplete the Ozone Layer was signed in 1987 and ratified two years later. With several subsequent amendments, the Protocol now controls (limits) the production and consumption of all major long-lived ODSs, which are ultimately emitted to the atmosphere. The atmospheric abundances of these species have responded to these controls; the stratospheric levels of chlorine and bromine peaked in the 1990s and are now slowly declining (e.g. Newman et al., 2007; Engel et al. 2018). In consequence, an increase ('recovery') of stratospheric ozone has been detected in the upper stratosphere and the Antarctic, although the signal is currently small and is difficult to separate from other atmospheric influences (e.g. Chipperfield et al., 2017). Nevertheless, the Protocol can therefore be considered on track in its aim of protecting the ozone layer from the effects of halogenated ODSs (see Section 3). A common measure of recovery is the date at which stratospheric ozone values are predicted to return to 1980 levels, before the occurrence of large depletion. This return will also be affected by factors other than ODSs, notably climate change (see Section 4). Models predict that this will occur around the middle of this century (e.g. Dhomse et al., 2018), although there are limitations using this simple measure of the timing of a specific event for quantifying the ongoing process of recovery (e.g. Pyle at al., 2022). Accordingly, the Montreal Protocol (MP) is arguably the most successful international environmental treaty to date. Recent discoveries related to increased emissions of controlled ODSs (Monztka et al., 2018) and uncontrolled short-lived halogenated source gases (e.g. Hossaini et al., 2017) had raised some concerns on the continued success of the treaty and the outlook for ozone recovery. However, the success of dealing with the CFC-11 issue (see Section 3.1) has demonstrated the resilience of the protocol, the effectiveness of its provisions, and the importance of continued vigilance regarding atmospheric trace gases.

This Opinion paper gives our personal view of some long-standing and recently emerging issues in ozone layer science. It is not a review of the subject; there are many excellent text books and the 4-yearly WMO/UNEP assessments (e.g. WMO, 2022) which serve that purpose. Section 2 gives a brief summary of ozone layer research, with emphasis on the contribution of Paul Crutzen, to whom selected papers in this issue are dedicated. Section 3 addresses the long-standing issue of ozone depletion driven by halogenated species. Section 4 discusses the impact increasing greenhouse gas loadings on stratospheric ozone and the new research areas of wildfire smoke and the expanding topic of volcanic impacts. Section 5 gives some thoughts on issues related to the availability of observations necessary to follow the evolution of the ozone layer and understand its changes. Section 6 discusses the range of modelling tools available, some further developments that are still needed, and how these tools can be best employed. Finally, an outlook is provided in Section 7.

## 2 A Century of Ozone Layer Research

Active research into stratospheric ozone dates back around 100 years. Dobson pioneered the detection and quantification of ozone in the stratosphere using a UV spectrometer (Dobson and Harrison, 1926) following earlier work by Fabry and Buisson (see historical summary in Brasseur, 2020). A theoretical model for creation of a stratospheric ozone layer, based solely on oxygen chemistry, was first proposed by Chapman (1930). This was based on the slow production and destruction of 'odd oxygen' ($Ox = O_3 + O(^3P)$) along with fast interconversion of $O_3$ and $O(^3P)$ within the Ox family. This oxygen-only model appeared to suffice until the 1960s when improved observations and laboratory measurements of key rate coefficients revealed a major quantitative discrepancy. The Chapman cycle included the only significant chemical source of Ox, i.e. photolysis of $O_2$, but ignored around 80% of stratospheric Ox loss via catalytic cycles that destroy ozone through reactions involving HOx (e.g. Nicolet, 1970), NOx and halogen radicals.

Here, as part of this special issue, we highlight the contribution of the late Paul Crutzen (1933 – 2021) to ozone layer science. For a comprehensive summary of his whole career please see Müller et al. (2022), and references therein, Solomon (2021a), Fishman et al. (2023) and Müller et al. (2023). Paul Crutzen started contributing to our understanding of the ozone layer very

early in his scientific career. In 1965, at Stockholm University, Crutzen helped visiting US scientist J.R. Blankenship to develop a numerical model of different forms of oxygen in the stratosphere, mesosphere and lower thermosphere. This marked the start of his scientific career and gave him his first paper (Blankenship and Crutzen, 1965). Following this work Crutzen chose to study for a PhD in stratospheric ozone as it appeared, at that time, to be a topic of "pure science related to natural processes" rather than one about human impact. Clearly, that situation later changed! In due course Crutzen submitted his PhD thesis 'On the photochemistry of ozone in the stratosphere and troposphere and pollution of the stratosphere by high-flying aircraft' to Stockholm University in May 1973.

In his PhD work (Crutzen 1970, 1971, 1972, 1973), Crutzen was the first to suggest that reactions catalysed by NO and $NO_2$ control the abundance of ozone in the middle stratosphere (around 25-35 km). This is summarised by the cycle:

$$NO + O_3 \rightarrow NO_2 + O_2$$

$$NO_2 + O(^3P) \rightarrow NO + O_2$$

Where the sum of NO and $NO_2$ is termed NOx. This discovery was a major achievement and helped to pave the way for a quantitative understanding of stratospheric ozone whereby catalytic cycles driven by radical species from various chemical families (HOx, NOx, Clx, Brx) are added to the original oxygen-only model of Chapman (1930). This work formed part of the basis for Crutzen being awarded the 1995 Nobel Prize for Chemistry jointly with Mario J. Molina and F. Sherwood Rowland "for their work in atmospheric chemistry, particularly concerning the formation and decomposition of ozone".

Prior to submitting his PhD thesis Crutzen spent two years (1969-1971) as a visitor to the University of Oxford. Here he developed his ideas on the importance of NOx in controlling ozone in order to address the issue of human-induced perturbations to the ozone layer caused by emissions from high flying supersonic transport (SST) aircraft. The debate on the atmospheric impacts of SST had begun in the early 1970s when it was envisaged that large fleets of around 500 aircraft such as the Anglo-French Concorde might be flown within the lower stratosphere (e.g. Johnston, 1971). Through his modelling work Crutzen was aware of inherent model uncertainties which prompted him to make the statement that the "minimum requirement is therefore that extensive supersonic air traffic should not take place in the stratosphere before reliable predictions can be made of the possible environmental consequences of such operations" (Crutzen, 1972). This is an insightful lesson that would be equally applicable to many other past and present areas of atmospheric science and therefore one well worth remembering.

During the 1970s Crutzen's scientific interests extended into other areas, though he did maintain a link with the stratosphere through the study of the impact of NOx produced from solar proton events on the ozone layer (Solomon and Crutzen, 1981). He also addressed the budget of stratospheric NOy (reactive odd nitrogen) from the perspective of surface sources, highlighting the human impact on stratospheric ozone of increased fertilizer and associated increased emissions of $N_2O$, the main source of stratospheric NOy (Crutzen and Ehalt, 1977). Following the same reasoning as for $N_2O$, he also worked out that the dominant non-volcanic source of stratospheric sulfur was the surface emissions of carbonyl sulfide (COS), the long-lived atmospheric sulfur compound (Crutzen, 1976).

Before his works on $N_2O$ and COS, Crutzen had also keenly followed the publication of seminal papers on chlorofluorocarbons (CFCs) (Molina and Rowland, 1974) and stratospheric chlorine (Stolarski and Cicerone, 1974), prompting him to publish a contribution on this topic (Crutzen, 1974). Similarly, following the surprise discovery of the Antarctic Ozone Hole (Farman et al., 1985) Crutzen was quick to think about the possible implications of co-condensation of $HNO_3$ and $H_2O$ (at temperatures above that at which pure ice clouds form) as a mechanism for widespread formation of polar stratospheric clouds (PSCs) and the initiation of key ozone-destroying halogen chemistry (Solomon et al., 1986) via heterogeneous reactions on PSCs (Crutzen and Arnold, 1986). In this way Crutzen made important scientific contributions to the early research into the causes of polar

ozone depletion. His later work on describing the epoch of the Anthropocene still has ongoing relevance to protecting the
Earth's ozone layer shield (see Solomon et al., 2021b). His multi-faceted scientific legacy for stratospheric ozone is assured.

## 3 Ozone Depletion and the Montreal Protocol

### 3.1 Montreal Protocol

The signing of the Montreal Protocol on Substances that Deplete the Ozone Layer in 1987 and its subsequent amendments
have had a major impact on the anthropogenic halogen source to the stratosphere. The Protocol now controls (limits) the
production and consumption of all major long-lived ODSs, which are ultimately emitted to the atmosphere. Controls on ODS
production have caused a net reduction in the emission and abundance of tropospheric source gases (**Figure 1a and b**) that
transport chlorine and bromine to the stratosphere.

A very important recent development in the Montreal Protocol was the inclusion of hydrofluorocarbons (HFCs) in the Kigali
Amendment of 2016 (WMO, 2018). HFCs do not contain any chlorine or bromine and hence do not lead directly to ozone
depletion. However, they are potent greenhouse gases and are only present in the atmosphere as replacements for CFCs and
HCFCs, hence the need to control these gases and to do so within the MP.

The majority of long-lived halocarbon source gases are now controlled by the Protocol. Further, or more rapid, reductions in
stratospheric chlorine (and bromine) would depend on extension of the Protocol to chlorinated very short-lived substances
(VSLS), defined as having an atmospheric lifetime of less than 6 months. The prime example of this is dichloromethane
($CH_2Cl_2$) (Hossaini et al., 2017) which is mainly of anthropogenic origin and, although largely removed in the troposphere,
does deliver a large fraction of the estimated $130 \pm 20$ pptv of VSLS chlorine to the stratosphere both directly (source gas
injection, SGI) and through decay products (product gas injection, PGI) (see WMO, 2022). Although this is only around 4%
of the current stratospheric chlorine loading, its contribution is expected to increase (Section 3.3).

The history of the MP since its signing in 1987 (and ratification in 1989) is one of continued success – as evidenced by the
decreasing loading of ODSs and stratospheric chlorine and bromine (WMO, 2022). Indeed, the former UN Secretary General,
Kofi Annan, described the Protocol to be not only "the most successful environmental treaty in history", but also "perhaps the
most successful international agreement to date" of any kind. However, that success appeared to be challenged for the first
time by the observation of an unexpected slowdown in the atmospheric CFC-11 decay (Montzka et al., 2018), which implied
renewed emissions. A large fraction (at least) of these emissions were traced to eastern China (Rigby et al., 2019). It must be
emphasised that this detection of apparent contravention of the MP was only possible through continued observations by the
distributed ground-based monitoring networks (see Section 5). Following this discovery, alarm was raised by policy makers
involved in the MP process that these renewed emissions could cause a delay in recovery of the ozone layer (e.g. Dhomse et
al., 2019). Extensions of these observations for a further three years (Montzka et al., 2021; Park et al., 2021) show that these
renewed emissions of CFC-11 appeared to have greatly declined. Therefore, we can argue that this episode has been further
evidence of the success of the MP and of the effective combination of monitoring observations, science and policy. We
emphasise that a key component of these interconnected activities is communicating ODS and ozone layer science and findings
to policy makers to guide future decision making to protect ozone and climate. Despite the undoubted progress in our
understanding of the atmospheric abundance of the major ODSs, some important and persistent uncertainties remain. In
particular, this is the case for carbon tetrachloride ($CCl_4$) (Sherry et al., 2018), which is produced in large quantities for
feedstock use (e.g. Chipperfield et al., 2020) and also has soil and oceanic sinks (e.g. Butler et al., 2016). It has proved
challenging to pin down the atmospheric budget of this species and explain the apparently slower atmospheric decay than
expected based on its estimated lifetime (e.g. Park et al., 2018). The continued observation of these controlled ODSs, and
further improved understanding of their atmospheric budgets, is important to ensure the ongoing success of the MP.

## 3.2 Ozone Recovery

The undoubted success of the Montreal Protocol in halting and turning around the increasing trend in stratospheric chlorine
and bromine is clearly expected to lead to ozone recovery, e.g. an increase in global ozone. However, the detection of ozone
recovery, and even maintaining consistency on the definition of what recovery is within the community, has proven difficult.
There is now a general consensus that recovery means 'recovery from the effects of depletion caused by halogen (chlorine and
bromine) species' (e.g. WMO, 2011). Stratospheric ozone amounts clearly depend on many other varying factors (e.g. solar
radiation, temperature, dynamics) which can also lead to an increase or a decrease in its concentration. These 'non-halogen'
influences need to be removed if the ozone recovery from halogens is to be quantified. Thus, recovery cannot generally be
detected directly from observations of ozone alone and a statistical or physical model is needed to isolate the effects of halogen
chemistry from other effects in the ozone evolution.

Given that recovery is from the effects of halogen-catalysed chemical depletion, the clearest signal of recovery might be
expected in regions where this chemistry exerts the strongest influence on ozone. Newchurch et al. (2003) first claimed the
detection of ozone recovery in the upper stratosphere where the classical ClO + O cycle (Molina and Rowland, 1974; Stolarski
and Cicerone 1974) has its maximum efficiency. In this region there are also non-halogen effects; in particular the contribution
of ozone increase from stratospheric cooling need to be removed, which is done by model attribution studies of the different
processes (noting that the stratospheric temperature changes can be due to both increased longwave cooling by $CO_2$ and
reduced shortwave heating by $O_3$ itself). It proved more elusive to detect recovery in the other atmospheric regions subject to
large halogen-catalysed loss - namely the polar lower stratospheres. Solomon et al. (2016) succeeded in detecting Antarctic
recovery (or 'healing') by focussing on the period of rapid chemical loss in September, rather than the period of lowest ozone
in October which is subject to saturation of the ozone loss and variability in breakdown of the polar vortex. As expected, the
larger interannual variability in Arctic ozone loss has made detection of any trends in this region difficult. However, using
long-term ground-based UV-visible observations, Pazmino et al. (2023) recently claimed some measure of Arctic ozone
recovery. These studies show that when searching for the signal of ozone recovery in a variable atmosphere it is important to
bear in mind that the different metrics used for the same phenomenon may indicate different behaviours for the recovery.

At extrapolar latitudes, observations confirm that the ozone decline in 1990s and earlier, caused by increasing atmospheric
concentrations of ODSs, has now transitioned to a slow ozone increase in both hemispheres (**Figure 2**, WMO (2022)). This is
consistent among the ground- and satellite-based measurements and chemistry-climate model simulations in the middle and
upper stratosphere, despite the larger variability of the ground-based measurements. This is apparent in the evolution of
observed and modelled annual mean deseasonalized ozone anomalies, relative to the 1998–2008 climatology for each
individual dataset in **Figure 2**, in the upper stratosphere (42 km or 2 hPa) and in the lower stratosphere (19 km or 70 hPa).
Upper stratospheric ozone anomalies averaged over 2017–2020 from most datasets are positive relative to the 1998–2008
average, which is consistent with expectations from the chemistry-climate model (CCM) simulations. In contrast, lower-
stratospheric ozone anomalies over 2017–2020 continue to be about the same as for the 1998–2008 average. Interestingly, in
2019 and 2020, stratospheric ozone values were lower than in previous years and below the level expected from model

simulations (Weber et al., 2020). The particularly low 2020 annual mean is the result of a very weak Brewer Dobson circulation (BDC) and a large and stable Antarctic ozone hole (Klekociuk et al., 2021; Weber et al., 2021). Such large interannual variability, driven by variations in meteorology and transport (e.g. Chipperfield et al., 2018), is typical for the lower stratosphere and limits our ability to drawing definite conclusions about long-term trends, especially for the mid-latitudes (30°–60°) in both hemispheres (see WMO, 2022). Evidently, longer observational time series should reduce the uncertainty due to this variability, again reinforcing the need for continued atmospheric monitoring.

While we can see that stratospheric halogen levels are decreasing, and therefore their impact on ozone is decreasing, there are a number of concerns about the extent and rate of ozone recovery. Clearly, ongoing emissions of chlorine and bromine from ODSs or VSLS that are not already accounted for will act to slow down this recovery (Sections 3.1 and 3.3). However, there are other factors which are not controlled by the MP and which may also lead to decreases in column ozone, ultimately the parameter of primary concern for protecting the biosphere. There are many studies (e.g. Ball et al., 2018, see also **Figure 2**) which point to an ongoing decrease in ozone in the mid-latitude lower stratosphere. This may be related to dynamical changes, which are predicted to decrease tropical column ozone in the future (Section 6). The model simulations of Chipperfield et al. (2018) supported this cause and showed a negligible impact of assumed trends on VSLS bromine and chlorine. In contrast, Villemayor et al. (2023) have suggested a role for the combined effects of chlorine, bromine and iodine VSLS acting together. This is a region where further work is needed to determine the extent of ozone depletion/recovery and to quantify its driving factors.

**3.3 Other Issues Related to Halogen Chemistry**

As noted in Section 2.2, VSLS deliver important amounts of chlorine and bromine to the stratosphere. VSLS bromine is largely of natural oceanic origin and contributes $5 \pm 2$ pptv to stratospheric bromine, which is around 27% of the total (WMO, 2022). There is currently no suggestion of a trend in this VSLS bromine contribution but this could potentially occur due to climate feedbacks on the strengths of the emission sources. In contrast, VSLS chlorine is largely of anthropogenic origin. Although the total VSLS chlorine injection of $130 \pm 20$ pptv is only 4% of the total stratospheric chlorine (WMO, 2022), it is showing a small increasing trend notably through increases in the atmospheric abundances of $CH_2Cl_2$ and $CHCl_3$ (e.g. Fang et al., 2019; Claxton et al., 2020). Far larger local stratospheric chlorine inputs from VSLS have recently been observed in regions where strong convection and emissions co-locate, notably the Asian Summer Monsoon (Adcock et al., 2020), pointing to the importance of observing chlorine species directly in the lower stratosphere.

Solomon et al. (1994) pointed out that iodine depletes ozone more efficiently than chlorine, and thus could be responsible for significant contribution to past and future ozone changes. However, there are still large uncertainties in the main gas- and condensed-phase iodine photochemical processes (see e.g. Saiz-Lopez et al., 2012; Feng et al. 2023) and observations of inorganic iodine (Iy) species in the upper troposphere – lower stratosphere (UTLS) are sparse. So far, only a few global 3-D models have included iodine chemistry (e.g., atmospheric chemistry-climate models such as CAM by Ordóñez et al., 2012; SOCOL-AERv2-I by Karagodin-Doyennel et al., 2021; WACCM by Cuevas et al., 2022; LMDZ-INCA by Caram et al., 2023; Chemical transport models MOZART by Youn et al., 2010; TOMCAT/SLIMCAT by Hossaini et al., 2015 and GEOS-Chem by Sherwen et al., 2016). These models have included the major sources of iodine from the ocean, including short-lived iodocarbons (e.g. $CH_3I$, $CH_2I_2$) and primary HOI and $I_2$ emissions (e.g., Carpenter, 2003; Jones et al., 2010, Saiz-Lopez et al., 2012; Carpenter et al., 2013). Recent measurements have indicated that up to $0.77 \pm 0.10$ parts per trillion by volume (pptv) total inorganic iodine reaches the stratosphere from ocean emissions (Koenig et al., 2020). Modelling studies have indicated that iodine may play an important role in stratospheric ozone depletion. However, large uncertainties remain over the

contribution of iodine to stratospheric ozone levels, ranging from a few percent reduction (e.g., Hossaini et al., 2015;
Karagodin-Doyennel et al., 2021) to 10% (Cuevas et al., 2022) and up to 30% (e.g., Ordóñez et al., 2012). Indeed, the
contribution of iodine could become more pronounced in the future (Cuevas et al., 2022; Villemayor et al., 2023) with the
decreasing amounts of stratospheric chlorine and bromine brought about by the Montreal Protocol (Feng et al., 2021).
It is worth pointing out that volcanoes are also a potentially significant source of halogens to the atmosphere (Bobrowski et
al., 2003; Pyle and Mather, 2009). Large halogen-rich eruptions could in principle inject large amounts of halogens, notably
bromine, directly into the stratosphere, causing massive ozone destruction (Kutterolf et al., 2013; Cadoux et al., 2015).
However, this phenomenon has not been observed during the current satellite era.

## 4 Other Challenges

The MP has been focused on reducing ozone depletion by anthropogenic halogens. However, there are other well-known
causes of global ozone perturbations, notably natural ones such as the 11-year solar variability (for which the recent solar cyle
23 showed decreased flux) and stratospheric sulfur injections by large volcanic eruptions (e.g. El Chichon in 1982 and Mt
Pinatubo in 1991) (WMO, 2022). So far, since the start of satellite observations around 1980, these natural factors have had a
relatively limited impact on global ozone and, unlike the anthropogenic halogen emissions, are only expected to cause short-
term (decadal timescale at most) fluctuations in stratospheric ozone.
Climate change represents a pressing and long-term issue for stratospheric ozone. The overall impacts of climate change
(largely driven by the increase in $CO_2$ levels) on stratospheric ozone are complex with uncertainties ranging from transport to
chemistry effects and their couplings (e.g. changes in the strength of the stratospheric BDC, changes in the tropospheric water
flux into the stratosphere, temperature-dependent chemistry effects, chemistry changes linked to the increasing levels in
stratospheric source gases such as $CH_4$ and $N_2O$ that are also major greenhouse gases) (e.g. WMO, 2018; 2022). Many of these
effects are coupled and some of the resulting stratospheric perturbations can, in return, influence the surface climate. For
example, the projected increasing speed of the stratospheric BDC will decrease column ozone in the tropics – a region which
has so far not been subject to substantial column depletion (Eyring et al., 2007). Increasing levels of $N_2O$ will lead to enhanced
NOx-catalysed ozone depletion in the middle atmosphere (Revell et al., 2012). The impact of increasing $CH_4$ is more complex;
it could lead to increased ozone depletion through increased HOx but less chlorine-catalysed depletion through deactivation
of Cl to HCl (Revell et al., 2012). These effects will increase as GHG levels increase but the details will depend on the relative
changes in $CO_2$, $N_2O$ and $CH_4$. Therefore, the chemical details of the different prescribed scenarios are important for the ozone
impact. Understanding and forecasting the effects of climate change on stratospheric ozone has been a major outstanding
challenge for several decades now and will remain one for years to come. While our knowledge of relevant atmospheric
processes will improve, there will remain the issue of uncertainty in GHG scenarios which are based on societal decisions.
More recently, other new challenges have emerged. The stratosphere contains aerosol particles which are mostly located in its
lower altitude region. This stratospheric aerosol load is usually dominated by supercooled sulfuric acid particles whose main
sources are stratospheric oxidation of volcanic $SO_2$ and of OCS, a long-lived sulfur species emitted at the surface (Crutzen,
1976). Sulfuric acid aerosols play an important role in stratospheric chemistry and in the radiative balance of the atmosphere,
notably when it is enhanced volcanically. They provide surfaces for key heterogeneous reactions (Hofmann and Solomon,
1989), cool the surface by scattering incident sunlight back to space and can heat the stratosphere by absorbing near-infrared
radiation (Stenchikov et al., 1998; Robock, 2000). Until quite recently, almost all the observed global enhancements in

stratospheric aerosols and resulting ozone perturbations were linked to sulfur injections by large volcanic eruptions (e.g. El Chichon in 1983, Mt Pinatubo in 1991). As the stratospheric aerosol variability appeared to be essentially driven by volcanic sulfur inputs, only sulfur-induced perturbations of stratospheric aerosols have usually been considered significant for the global stratosphere and climate. This focus on sulfur has also led to the development of sophisticated stratospheric sulfate aerosol microphysical modules which are now implemented in several global climate models (e.g. Zanchettin et al., 2016). These models are able to reproduce observed features of the stratospheric aerosol layer rather well, especially the large enhancements by volcanic eruptions (Zanchettin et al., 2022) and associated ozone losses (e.g. Bekki and Pyle, 1994; Mills et al., 2017). These models are also used to assess the impacts of other stratospheric sulfur injections on stratospheric ozone, for example from aircraft or potential stratospheric geoengineering (Pitari et al., 2014).

Aircraft measurements in the lowermost stratosphere have already revealed that the nature and composition of stratospheric aerosols are more variable and complex than assumed in most stratospheric aerosol-climate models. These usually only consider sulfur and ignore the substantial components of meteoritic and organic material, dust, and metallic particles from space activities (Murphy et al., 2014; Martinsson et al., 2019; Murphy et al., 2021; Schneider et al., 2021; Murphy et al., 2023). Two recent events have further challenged the dominant view that sulfur is the only aerosol component relevant for the global stratosphere, ozone layer and climate. The first event was the massive Australian wildfires at the turn of 2020, the so-called Australian New Year's (ANY) event (Khaykin et al., 2020; Peterson et al., 2021); the second event was the eruption of the Hunga Tonga – Hunga Ha`apai (HTHH) volcano in January 2022 (Carr et al., 2022; Zuo et al., 2022). The nature and magnitude of the various stratospheric impacts of these two events have been unexpected and sometimes unprecedented in the historical records. After extensive research on the stratosphere since the discovery of the Antarctic Ozone Hole in 1985, these two recent events represent extreme but valuable testbeds of our understanding and modelling of stratospheric physics and chemistry.

## 4.1 Australian Wildfires

### 4.1.1 Injections of Carbonaceous Particles and Resulting Aerosol Changes

Wildfires can trigger the formation of pyrocumulonimbus (PyroCb) towers that can, depending on the meteorological conditions and intensity of the fires, rise high enough to transport biomass-burning material into the UTLS (Peterson et al., 2018). The Australian 'Black Summer' wildfires of 2019–2020 were exceptional in terms of scale, intensity and stratospheric impacts according to historical records (Damany-Pearce et al., 2022). The strongest set of PyroCb outbreaks (ANY) occurred at the turn of 2020, injecting massive amounts of gaseous and particulate biomass-burning products above the tropopause. For instance, ~1 Tg of carbonaceous aerosols and ~25 Tg of $H_2O$ were released into the lower stratosphere during the main ANY event (Khaykin et al., 2020; Damany-Pearce et al., 2022; Ohneiser et al., 2022), resulting in a sharp increase in global stratospheric aerosol optical depth (SAOD). The rise in SAOD was comparable to the increases produced by the strongest volcanic eruptions since Mt Pinatubo in 1991, namely Calbuco in 2015 and Raikoke in 2019 (see **Figure 3**). Stratospheric aerosol levels remained enhanced in the Southern Hemisphere throughout 2020. Note that the radiative properties and heterogeneous chemistry of carbonaceous aerosols are different from those of sulfate aerosols (Yu et al., 2022). As a result, the impacts of ANY aerosols on the stratosphere and surface climate are expected to differ from those of volcanic sulfate aerosols.

## 4.1.2 Gaseous Composition Changes

ANY stratospheric aerosol changes were accompanied by very unusual large-scale perturbations in gaseous composition. For example, in the months following the ANY aerosol dispersion, unexpected partitioning between radicals and reservoir species in the chlorine and nitrogen families were observed at southern mid-latitudes at relatively warm stratospheric temperatures (Santee et al., 2022). The main stratospheric chlorine reservoir species HCl was found to be largely depleted while the other chlorine reservoir, $ClONO_2$, and the ozone–destroying chlorine radical ClO, were enhanced. The anomalous partitioning is somewhat reminiscent of the effects of ozone-depleting heterogeneous chemistry on other stratospheric aerosols (sulfuric acid particles, PSCs) and was probably initiated by some heterogeneous processing on ANY particles (Bernarth et al., 2022; Solomon et al., 2023). Overall, the enhanced ClO concentrations likely caused some, albeit weak, chemical ozone depletion. A mini ozone hole (depletion of up to 100 DU) was also apparent early on within the largest plume vortex (Section 4.1.3; Khaykin et al., 2020) and the Antarctic ozone hole was particularly long-lasting in 2020 (Klekociuk et al., 2022). Several aerosol-driven mechanisms have been proposed to explain these ozone changes, invoking changes in stratospheric dynamics and/or heterogeneous chemistry (e.g. Ansmann et al., 2022).

It has to be stressed that, at this stage, we do not know exactly the physical state (e.g. liquid, glassy, solid) and composition of such wildfire particles in the conditions prevailing in the stratosphere, all the more so when internally mixed with sulfuric acid (Solomon et al., 2023). As a result, the types and rates of heterogeneous reactions occurring on them can only be hypothesised. Further laboratory studies and, as importantly, detailed chemical composition measurements are certainly our best means to characterise unequivocally the physico-chemistry of these aerosols.

## 4.1.3 Dynamics and Radiative Forcing

In the early phase when aerosol concentrations within the ANY plumes were extremely high, the intense solar heating by the highly absorptive ANY aerosol plumes led to very peculiar dynamical feedbacks and the formation of self-maintained anticyclonic vortices. This included one with a size of ~1000 km, which contained extremely high concentrations of wildfire gases and aerosols. The massive and remarkably compact vortex persisted for several months while rising diabatically to ~35 km (Khaykin et al., 2020). The aerosol lofting opposed the effect of gravitational settling, prolonging the residence time of ANY aerosols in the stratosphere. Interestingly, after the discovery of heating and self-lofting by ANY carbonaceous aerosols, an analysis of high-resolution satellite observations has showed that the Raikoke volcanic eruption in 2019 also generated a stratospheric anticyclonic vortex which rose to 27 km and persisted for more than 3 months (Khaykin et al., 2022). Since sulfate aerosols absorb radiation only weakly, the heating must have been generated by absorption from another volcanic aerosol component, likely to be volcanic ash. Currently, most stratospheric aerosol models only consider sulfate aerosols and hence cannot reproduce the observed dynamical confinement and ascent of concentrated carbonaceous plumes or ash-rich plumes and hence the extended residence time in the stratosphere. Once the ANY aerosol plumes were dispersed and spread, the aerosol heating led to a pronounced large-scale warming of the southern lower stratosphere (Stocker et al., 2021; Damany-Pearce et al., 2022) which was stronger than any warmings from recent volcanic eruptions.

The climate forcing by ANY aerosols is more difficult to estimate than the forcing by sulfate aerosols. Sulfate aerosols cool the surface by efficiently scattering incoming sunlight back to space and this effect readily dominates the surface-warming tendency from their absorption of longwave radiation. Carbonaceous aerosols not only scatter solar radiation but also absorb it, and this absorption is strongly dependent on the aerosol composition. ANY aerosols are thought to have been mostly composed of a small fraction of black carbon (BC, soot-like component) and a vastly dominant fraction of organic material (OM, including the so-called brown carbon (BrC) component) (Liu et al., 2022). BC absorbs across the entire solar spectrum

and hence is by far the most efficient source of heating. Most OM compounds absorb strongly in the IR and UV wavelengths, but are relatively transparent in the visible and near-IR wavelengths. This is not the case for BrC which can also absorb in the blue and near-UV spectral regions, albeit with a much weaker efficiency than BC (Laskin et al., 2015; Yu et al., 2021). Given the poor observational constraints on the composition, physical and mixing state, and size distribution of ANY carbonaceous aerosols (all key parameters of aerosol radiative properties), the radiative impact of ANY aerosol remains as difficult to assess as their heterogeneous chemistry. Estimations of ANY aerosol surface radiative forcing (RF) vary from negligible to about -1 $Wm^{-2}$; this range can be compared to the RF of small to moderate volcanic eruptions during the last 3 decades, estimated at between $-0.1$ and $-0.2\,Wm^{-2}$ (Sellitto et al., 2022a; Liu et al., 2022). An additional complication in the ANY RF estimation is the effect of the aerosol-driven stratospheric warming on the longwave radiation budget (Liu et al., 2022).

It is worth pointing out that, as global surface warming intensifies, massive wildfires and associated pyro-convective injections of carbonaceous particles in the stratosphere are expected to become more frequent. Pyro-convection could turn into a significant source of large-scale perturbations of stratospheric aerosols, ozone, and climate. Therefore, it might be necessary to account for stratospheric wildfire particle processes in CCMs and comprehensive Earth system models (ESMs) in the future.

## 4.2 Hunga Tonga – Hunga Ha`apai Volcanic Eruption of January 2022

### 4.2.1 Injection of $H_2O$ and Sulfur

The eruption of the Hunga Tunga – Hunga Ha'apai volcano with an underwater caldera occurred on January 15th 2022. Several features of this eruption were unique in the record of stratospheric observations. First, it generated a very powerful blast that injected volcanic material up to an altitude of nearly 58 km (Proud et al., 2022; Carr et al., 2022). A volcanic plume reaching the lower mesosphere was barely conceivable until this event, especially when the plume of the Mt Pinatubo eruption in 1991 with an explosivity index larger than the HTHH eruption reached at most an altitude of ~40 km (McCormick et al., 1995). Second, the HTHH eruption injected a very small amount of $SO_2$ (0.4-0.5 Tg) but a massive quantity of $H_2O$, between 120 and 150 Tg (Carn et al., 2022; Millan et al., 2022; Xu et al., 2022; Khaykin et al., 2022), into the middle atmosphere, resulting in very large increases in stratospheric water vapour (see **Figure 4**). Again, such a volcanic emission scenario had not been generally considered previously. $H_2O$ isotopic ratio data strongly indicate that sea water was a major source of stratospheric hydration by the HTHH eruption (Khaykin et al., 2022), which is consistent with the high concentrations of sea salts found in HTHH tephra (volcanic ash) collected shortly after deposition at the surface (Colombier et al., 2023).

### 4.2.2 $H_2O$ and Sulfate Aerosol Changes

The HTHH eruption increased the mean global stratospheric water content by approximately 10%, which is unprecedented in the entire observational record dating back to 1985. Note that, as there are no significant sinks of $H_2O$ within the stratosphere, this excess $H_2O$ should persist at least several years during which time the water vapour is slowly transport to the troposphere. In contrast, volcanic sulfate particles have a shorter residence time in the stratosphere, with typically an e-folding decay time of a year, because of the effect of gravitational sedimentation. This difference between the gaseous and aerosol components has led to an increasing vertical decoupling of the HTHH enhanced water vapour and aerosol layers in the stratosphere (Millan et al., 2022; Khaykin et al., 2022).

Most of the HTHH $SO_2$ was oxidised to sulfate aerosols within a month because of the $H_2O$-driven OH enhancement (Zhu et al., 2022). The SAOD (averaged between 60°S and 60°N above 380 K) increased rapidly and reached a peak 5 months after the eruption (Khaykin et al., 2022). Surprisingly, the magnitude of the SAOD increment did not follow at all the common relationship between SAOD and volcanic $SO_2$ mass injected in the stratosphere. For instance, the HTHH SAOD enhancement, which easily outweighed all the volcanic and wildfire aerosol perturbations in the last three decades, exceeded by a factor 4 the SAOD peak caused by 2015 Calbuco eruption that injected roughly the same amount of sulphur, and by a factor 2 the SAOD peak caused by the 2019 Raikoke eruption that injected two times more $SO_2$ than the HTHH eruption (Khaykin et al., 2022). This unexpected increase in SAOD in the case of the HTHH could not be linked to the possible presence of volcanic ash because that was apparently removed within days after the eruption and, according to satellite data, HTHH aerosols were essentially liquid sulfate droplets (Legras et al., 2022; Bernath et al., 2023). The highly enhanced SAOD must have been due to the excess humidity in the stratosphere, possibly through aerosol hygroscopic growth or coagulation. Indeed, in sulfate aerosol microphysical model simulations of the HTHH eruption, the SAOD generated by a ~0.4 Tg $SO_2$ injection is approximately doubled by the co-injection of 150 Tg of water (Zhu et al., 2022). Nonetheless, the model still underestimates the observed SAOD by a factor 2, suggesting that the effect of water vapour on sulfate aerosols is yet not fully understood or that the HTHH aerosols were not just composed of sulfuric acid and water, possibly with sea salts affecting the aerosol hygroscopicity.

Satellite observations of trace gases have also provided strong evidence for heterogeneous chemical processing on HTHH aerosols with unambiguous signatures of substantial chlorine and nitrogen repartitioning in the regions of aerosol enhancements almost immediately after the eruption (Santee et al., 2023). Model simulations indicate that stratospheric ozone has been significantly impacted by the eruption through not only heterogeneous chemistry but also other chemical and dynamical mechanisms (e.g. $H_2O$-enhanced gas-phase radical chemistry, and circulation changes) (Lu et al., 2023).

### 4.2.3 Dynamics and Radiative Forcing

The first radiative forcing (RF) model calculations for HTHH took only into account the sulfur injection, ignoring the water injection, and, as expected, concluded that the HTHH sulfate aerosols would slightly cool the surface (Zuo et al., 2022). However, enhancements in lower stratospheric $H_2O$ and sulfate aerosols generally have opposite radiative impacts. A $H_2O$ increment tends to cool the stratosphere and warm the surface while a sulfate aerosol increment tends to warm the stratosphere and cool the surface. The water vapour content within the HTHH plume was initially so high that the $H_2O$ radiative cooling led to a descent of the volcanic plume during the first weeks after the eruption (Sellitto et al., 2022b). After this initial phase, negative temperature anomalies were found to be correlated with $H_2O$-rich layers (Schoeberl et al., 2022). The decrease in global temperatures was rather extreme in the mid-stratosphere during 2022, deviating markedly from all the previous 42 years of meteorological data (Coy et al., 2022). The sign of the stratospheric temperature response confirms that the $H_2O$ cooling clearly dominated the sulfate aerosol warming in the stratosphere. These temperature perturbations were also accompanied by circulation adjustments.

The effect of the HTHH event on surface climate is not as clear-cut as in the stratosphere. RF model calculations suggest that ultimately the eruption warmed the surface; i.e. that the $H_2O$ warming was slightly dominant over sulfate cooling (Sellitto et al., 2022b; Jenkins et al., 2023).

**5 Maintaining Observational Capacity**

Our understanding of the ozone layer, and of the processes which control its evolution including those outlined here, depends on the availability of high-quality observations. In recent years we have benefitted from a wealth of observations from instruments in ground-based networks and on balloon, aircraft and satellite platforms. However, there are several indications that future progress will be impeded by fewer observations in the future.

**5.1 Gaseous Composition**

Several currently operational spaceborne instruments are well beyond their design lifetimes, and some are scheduled to be decommissioned in the next few years. Instruments whose data have been cited above or regularly used as part of the 4-yearly WMO/UNEP Ozone Assessments (e.g. WMO, 2022) will likely cease operations within the next few years, including the Aura Microwave Limb Sounder (MLS), the SciSat Atmospheric Chemistry Experiment Fourier Transform Spectrometer (ACE-FTS), the Odin Optical Spectrograph and Infrared Imager System (OSIRIS), and the Odin Sub-Millimetre Radiometer (SMR). With the loss of these current limb-viewing capabilities, vertically resolved global measurements of many trace gases relevant for studies of stratospheric chemistry and dynamics will no longer be available. These trace gases include ozone-destroying reactive (ClO) and reservoir (HCl, $ClONO_2$) chlorine species, water vapour, nitric acid ($HNO_3$), and long-lived tracers of transport (e.g., nitrous oxide, $N_2O$; methane, $CH_4$; carbon monoxide, CO). As noted in WMO (2022), the 2021 Report of the Ozone Research Managers of the Parties to the Vienna Convention (ORM, 2021a) identified the need to "continue limb emission and infrared solar occultation observations from space" that are "necessary for global vertical profiles of many ozone and climate-related trace gases" as one of the "key systematic observations recommendations." Indeed, the impending loss of these measurements, many of which have been taken continuously over the last several decades, will hamper our ability to appreciate and address key gaps and uncertainties in our understanding of stratospheric ozone depletion, including the lack of emergence of a clear signature of recovery in the Arctic, the influences of volcanic and wildfire emissions, the role of VSLS, and the impact of strengthening of the Brewer-Dobson circulation. Ultimately, this will risk weakening the scientific framework of the Montreal Protocol including the decision-making process. It may take many years for the next generation of improved limb sounders to become operational and provide us with the observational capacity that has been so essential to understanding the evolution of the ozone layer over the past three decades. For example, the novel, high resolution Changing Atmosphere Infra-Red Tomography Explorer (CAIRT) (https://www.cairt.eu/) is currently a candidate mission for the European Space Agency Earth Explorer 11 mission but, if selected, will not start operating before 2033 at best.

Ground-based networks have also proved essential for the ozone layer research. Examples are the Network for the Detection of Atmospheric Composition Change (NDACC, De Mazière et al., 2018), and the National Oceanic and Atmospheric Administration (NOAA, e.g. Montzka et al., 2018) and Advanced Global Atmospheric Gases Experiment (AGAGE, e.g. Rigby et al., 2019) surface networks. While these networks have an important monitoring function, the data acquired have proved central to trend analyses, to the validation of satellite measurements, and to the identification of many of the new scientific challenges discussed here. The benefit of these data sets increases greatly as the time series extend so that longer term variations can be characterised and studied. Therefore, it is crucial to maintain their continuity as discussed, for example, in ORM (2021a, b).

## 5.2 Aerosol Composition

As for trace gases, the much reduced availability of satellite limb-viewing observations in the future is concern for research on stratospheric aerosols, an important driver of stratospheric ozone. Several spaceborne instruments arriving towards the end of their lifetimes have been providing critical information on stratospheric aerosol properties. It is not ideal to reduce stratospheric aerosol observations with their global coverage, especially when it is becoming increasingly clear that large-scale ozone perturbations from stratospheric aerosol changes are not limited to volcanic sulfur injections. The chemical composition of stratospheric aerosols is more variable and complex than often assumed. In addition to sulfuric acid and water, stratospheric aerosols contain significant fractions of carbonaceous, meteoritic, and space activity material but the impacts of some of these components on stratospheric ozone are more or less unknown. As a result of poor observational constraints, large uncertainties pertain to the sources, size distribution, heterogeneous chemical reactivity, possibly polar stratospheric cloud activation ability, or/and radiative properties of these components. This incomplete knowledge hinders our ability to foresee the state of the ozone layer under the effect of a range of potential aerosol perturbations such as massive wildfires expected to be favoured by global warming, the anticipated increase in space activities within the next few decades (e.g. Ryan et al., 2022), the impact of meteoritic particles (Plane et al., 2023) or stratospheric geoengineering (i.e. deliberate injection aerosols or/and gaseous precursors in the stratosphere in order to counteract climate warming (e.g. Tilmes et al., 2022)).

It is worth pointing out that satellite observations cannot alone constrain unambiguously key aerosol parameters, in particular chemical composition and size distribution. Satellite data have to be confronted and combined with in-situ detailed composition and size measurements from balloon and aircraft; in addition, laboratory studies help to characterise the primary processes relevant to the aerosol physico-chemistry (Burkholder et al., 2017). All these types of measurements are needed to advance our understanding of stratospheric aerosol processes and impacts, and thus improve their representations in models.

## 6 Chemistry-Climate Modelling and Ozone Projections

Our understanding of the chemical, dynamic and radiative processes and of their couplings which control stratospheric ozone is encapsulated in mathematical form in numerical models. These models are powerful tools in tackling a range of scientific and societal challenges. Obviously, they can only include known processes (as the surprise discovery of the Antarctic ozone hole clearly demonstrated) and even for these there can be significant uncertainties. Overall, progress in our understanding of the ozone layer will depend on the improvement and careful application of a hierarchy of models from detailed chemical-aerosol box models, through 3-D chemical transport models (CTMs) to complex chemistry-climate models (CCMs).

Computationally inexpensive 3-D CTMs will continue to play an important role in interpreting observations on a range of spatial and temporal scales, testing our understanding and developing parameterisations for new processes. These models contain detailed chemistry-aerosol schemes but are forced by off-line meteorological analyses making them ideal tools for comparing with observations and for many sensitivity studies. CCMs are needed to study chemical-radiative-dynamical interactions but these models are relatively very computationally expensive to run. Continuing advances in computing resources allow ever more complex processes to be added which can help understanding of feedback pathways but can mean that simulations are often at the limit of what is practical. For example, the stratospheric impact of halogenated VSLS will ideally require detailed tropospheric chemistry in order to accurately model the transport of product gases to the stratosphere. This 'whole atmosphere' chemistry is also desirable for many other reasons, but it adds to the cost of all stratospheric simulations, and to the amount of model output generated. A set of ensemble CCM simulations (needed to characterise the model internal variability) can take many months of real time even on a powerful High Performance Computing (HPC) system.

Moreover, the costs increase greatly as other modules, such as ocean, cryosphere and biosphere, are added to build a full ESM. Within an ESM framework there is, we think, a danger that the treatment of the stratosphere is simplified to such an extent that the model will not capture many of the important processes discussed above (e.g. VSLS, wildfire smoke) and thus will not produce the best estimate for processes such as ozone layer recovery. For example, the standard UKESM (Archibald et al, 2020) only treats three ODSs (2 CFCs and $CH_3Br$) with other simplifications for PSCs. In practice, other versions of the ESM may be available (in effect a 'CCM' if other modules are not used to save time) but these will likely not be used for flagship climate simulations in major international assessments. This is important not only for simulating the stratosphere itself but because changes in the stratosphere are known to exert important impacts on the troposphere and the surface (e.g. Thompson et al., 2011).

Given the computational challenges simulations with CCMs (and ESMs) need to be planned carefully. Results from any given model will have various causes of uncertainty: (1) internal variability; (2) structural uncertainty – related to the model grid and parameterisations used to represent known processes and (3) scenario uncertainty – related for example to the ODS and GHG scenarios used to force the model. To address (1) each CCM needs to perform an ensemble of simulations. To address (2) a selection of models are needed to perform a given experiment in order to obtain a robust result (in the sense that the result is not, or at least only weakly, model-dependent). To address (3) the models must be computationally cheap enough to simulate a range of possible scenarios. For example, as discussed above, an important use of CCMs is to predict recovery of the ozone layer from chlorine and bromine-catalysed loss, and the dependence of that recovery on climate change. These results are obtained from projects such as the Chemistry-Climate Modelling Initiative (CCMI, https://igacproject.org/activities/CCMI) and feed into the WMO/UNEP Assessments. It is important that the participating models have been thoroughly evaluated and that they perform sufficient experiments (with ensemble members). For example, as noted by Dhomse et al. (2018), robust estimates of sensitivity to GHG scenarios are better achieved when all (well evaluated) models perform all experiments and these results from around 20 models fed into the projections used in WMO (2018) (see **Figure 1c and d**). In comparison, projections used in WMO (2022) were based on only 5 or 6 models (depending on region) and from simulations that were performed for the wide-ranging Coupled Model Intercomparison Project 6 (CMIP6, https://pcmdi.llnl.gov/CMIP6/) which were not focussed on the stratosphere. While all models used for assessment purposes should ideally have comprehensive stratospheric processes, their projections of ozone recovery will always have some caveats. Clearly the models cannot contain unknown processes – and the recent example of chlorine activation on wildfire smoke particles (Section 4.1) is one example. Even for known processes, we do not know how external forcings, such as volcanic eruptions, will vary. Therefore, there will always be an important role for additional, focussed studies of chemistry-climate interactions and projections outside of the main assessment process in order to explore detailed interactions and accommodate new knowledge.

Given the increasing computational cost of the CCM/ESM simulations then it is desirable that other approaches are used to update projections of ozone layer recovery which do not depend on extensive new sets of model runs. A commonly used metric is the 'ozone return date' (see **Figure 1c and d**). This is the date at which modelled ozone levels return to a reference value, which is often taken to be 1960 or 1980. These return dates are, for example, typically around 2040 for global mean column ozone and 2066 for the Antarctic in October, but with large uncertainty due to e.g. GHG scenarios (WMO, 2022). Although an apparently simple metric, there are a number of obvious shortcomings with return dates. The return date measures recovery as a single event and does not take account the trajectory of ozone prior to that date, noting that the impact of increased surface UV will depend on time history of ozone depletion. Furthermore, small shifts in the extent of ozone depletion around the return date can cause a large 'delay' in when ozone recovery is deemed to have occurred. Performing simulations to update these estimates is also expensive with possibly only a small benefit if the ODS and GHG scenarios have only changed slightly.

Therefore, alternative approaches should be investigated for estimating, for example, the dependence of the ozone return date
on the chlorine and bromine return dates, and the sensitivity of this to different GHGs.

Recently, Pyle et al. (2022) proposed the Integrated Ozone Depletion (IOD) metric and showed how it applies to similar long-
lived ODSs. IOD is an absolute measure of the time-integrated column ozone depletion for different halocarbon scenarios
which, for long-lived ODSs, reduces to a simple empirical formula with a model-derived scaling factor. As noted above,
application of ODPs to VSLS depends on the distribution of the surface emissions, which leads to a range of IOD values.
Because VSLS can cause ozone changes in the troposphere, Zhang et al. (2020) proposed the use of 'stratospheric ODP'
(SODP) as a simpler and more direct measure of only stratospheric column changes. Further work from the modelling
community is needed to derive a robust range of (S)ODPs for VSLS, and to also extend the work of Pyle et al. (2022) to
investigate how to apply the IOD metric to VSLS. In particular, we need to test the sensitivity of simulated ozone depletion to
emissions (i.e. IOD scaling factor) in a range of models.

**7 Future Outlook**
This Opinion article demonstrates that after 100 years of research, and nearly 4 decades after the discovery of the Antarctic
ozone hole, the stratospheric ozone layer is still producing surprises and new research challenges. Clearly we cannot lower our
guard on this global environmental issue. The great progress that we have made in ozone layer science has been achieved
through the combination of laboratory studies, observations from a range of platforms, and modelling. All of these components
are essential for continued progress in research and policymaking concerning the preservation of the ozone layer.

Our reflections of the long-standing and new challenges presented in this paper can, we think, be summarised in the following
overarching research needs:
• Maintaining and expanding the observational monitoring networks to ensure compliance with the Montreal Protocol
for the controlled gases and to understand the distribution and emissions of important uncontrolled gases. This
monitoring should cover the important ODSs, the replacements of the ODSs and VSLS, and be of high enough
coverage that emissions can be traced to specific regional sources.
• Addressing the critical issue of the impending satellite gap in observational capacity of the stratosphere which will
greatly reduce our ability to study processes globally. In order to understand changes in stratospheric ozone, for
example to track recovery or understand new perturbations, we need height-resolved profiles of related chemical and
aerosol species. Other targeted observational campaigns from aircraft and balloons in the low-mid stratosphere are
critical for increasing the observational database to more species and providing observations at high spatial resolution.
• Supporting the development, testing and application of process models. This will also require relevant laboratory
studies to measure key parameters. CTMs will continue to be important tools to test understanding and interpret
observations. This development can feed into the chemistry-aerosol modules used in more complex CCMs.
• Ensuring that ESMs being developed worldwide treat the stratosphere in sufficient detail. Use of ESMs for assessment
simulations should be based on well-tested models, a sufficiently large number of ensemble members to account for
model internal variability and include provision for a range of scenarios and sensitivity runs. If the full ESM is too
costly for this then regular CCMs should be used. New metrics need to be explored, e.g. to quantify ozone recovery,
which provide direct measure of the impact of the process being considered and to reduce the need for a large number
of repeated expensive model runs as external forcings change only slightly.

Our personal experience has also convinced us of the great importance of collaborative international programmes and
campaigns which have been truly instrumental in advancing our knowledge on the topic. Ultimately, society's interest in the
ozone layer is due to the impact of ozone depletion on surface UV and climate. As this article has shown, although the ozone
layer is demonstrating recovery from the effects of long-lived ODSs, other processes such as uncontrolled short-lived species,
changing dynamics, and wildfire smoke, could cause further perturbations.
In summary, study of the stratospheric ozone layer continues to uncover novel research challenges and to reveal yet more
processes and mechanisms that can perturb this essential component of the Earth system. Global monitoring of stratospheric
ozone and of its gaseous and particulate drivers, combined with numerical modelling, remains absolutely vital if these
challenges are to be met. This is a prerequisite for reliable future projections.

**List of Acronyms**

ANY – Australia New Year

CCM – Chemistry-climate model

CMIP – Coupled Model Intercomparison Project

CTM – Chemical Transport Model

EECl – Equivalent effective chlorine

ESM – Earth system model

GHG – Greenhouse gas

HTHH – Hunga Tunga – Hunga Ha'apai

IOD – Integrated ozone depletion

IR - Infrared

MP – Montreal Protocol

ODP – Ozone depletion potential

ODS – Ozone depleting substance

PGI – Product gas injection

PSC – Polar stratospheric cloud

RF – Radiative forcing

SAOD – Stratospheric aerosol optical depth

SGI – Source gas injection

SODP – Stratospheric ODP

UTLS – Upper troposphere – lower stratosphere

UV – Ultraviolet

VSLS – Very short-lived substance

WMO – World Meteorological Organisation

**Author Contributions**

Both authors contributed to the writing of this article.

**Competing Interests**

The authors declare that they have no conflicts of interest.

**Acknowledgements**

MPC and SB thank numerous colleagues for past discussions on the topics described here. In particular, MPC thanks the co-lead authors of the recent WMO assessment ozone chapters: Michelle Santee, Birgit Hassler and Paul Young. MPC also thanks Ryan Hossaini, Sandip Dhomse and Xin Zhou for specific comments. SB thanks Sergey Khaykin, Gwenael Berthet, and Jean-Paul Vernier. MPC is grateful for support through the NERC LSO3 (NE/V011863/1) and ESA OREGANO (contract No. 4000137112/22/I-A) projects. SB thanks the support of the French Agence Nationale de la Recherche (ANR) PyroStrat project (Grant No: 21-CE01-0028-01) and Centre National d'Etudes Spatiales (CNES) BeSAFE project. We thank the two anonymous reviewers for their constructive criticism which has improved the final version of this paper.

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

**Figures**

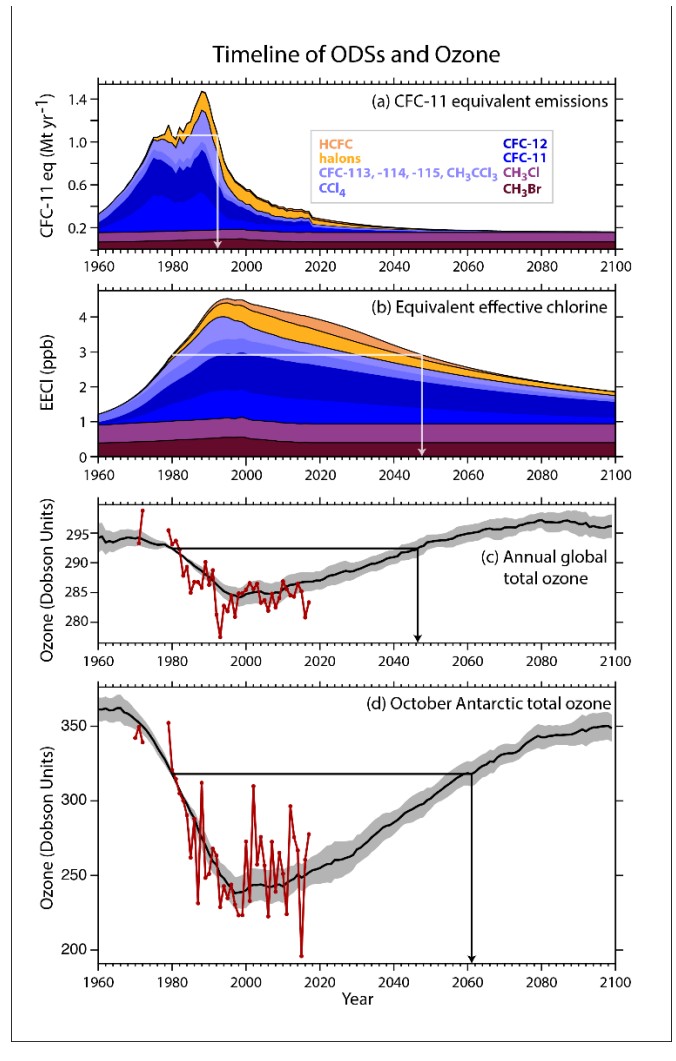

**Figure 1**: (a) CFC-11-equivalent emissions (Mt yr$^{-1}$) for major ozone depleting substances (ODSs) between 1960 and 2100 based on historical or projected emissions scaled by each the ozone depletion potential (ODP) of each species. (b) Past observations and projections of the equivalent effective chlorine (EECl; total chlorine + 65 × total bromine at surface, ppb) for the major ODSs. After the signing of the Montreal Protocol and subsequent phaseout of many long-lived ODSs, the EECl began to decline and is expected to return to 1980 levels by around 2050, as indicated by the horizontal and vertical dashed lines. Note that more recent estimates of EECl would give a slightly later return date (WMO, 2022). (c and d) Measured (red line) and predicted (black line, with uncertainty shown as grey shading) annual global (panel c) and October Antarctic (panel d) column ozone (Dobson units) between 1960 and 2100. In these simulations the Antarctic ozone layer is expected to return to 1980 levels around 2061, around a decade later than the EECl (horizontal and vertical dashed lines). CFC, chlorofluorocarbon; HCFC, hydrochlorofluorocarbon. Note that this Antarctic October return date is slightly earlier than the most recent estimate given in WMO (2022) but still within the model uncertainty range. Figure taken without change from Figure ES-1 in WMO (2018) (see also Chipperfield et al. (2020)).

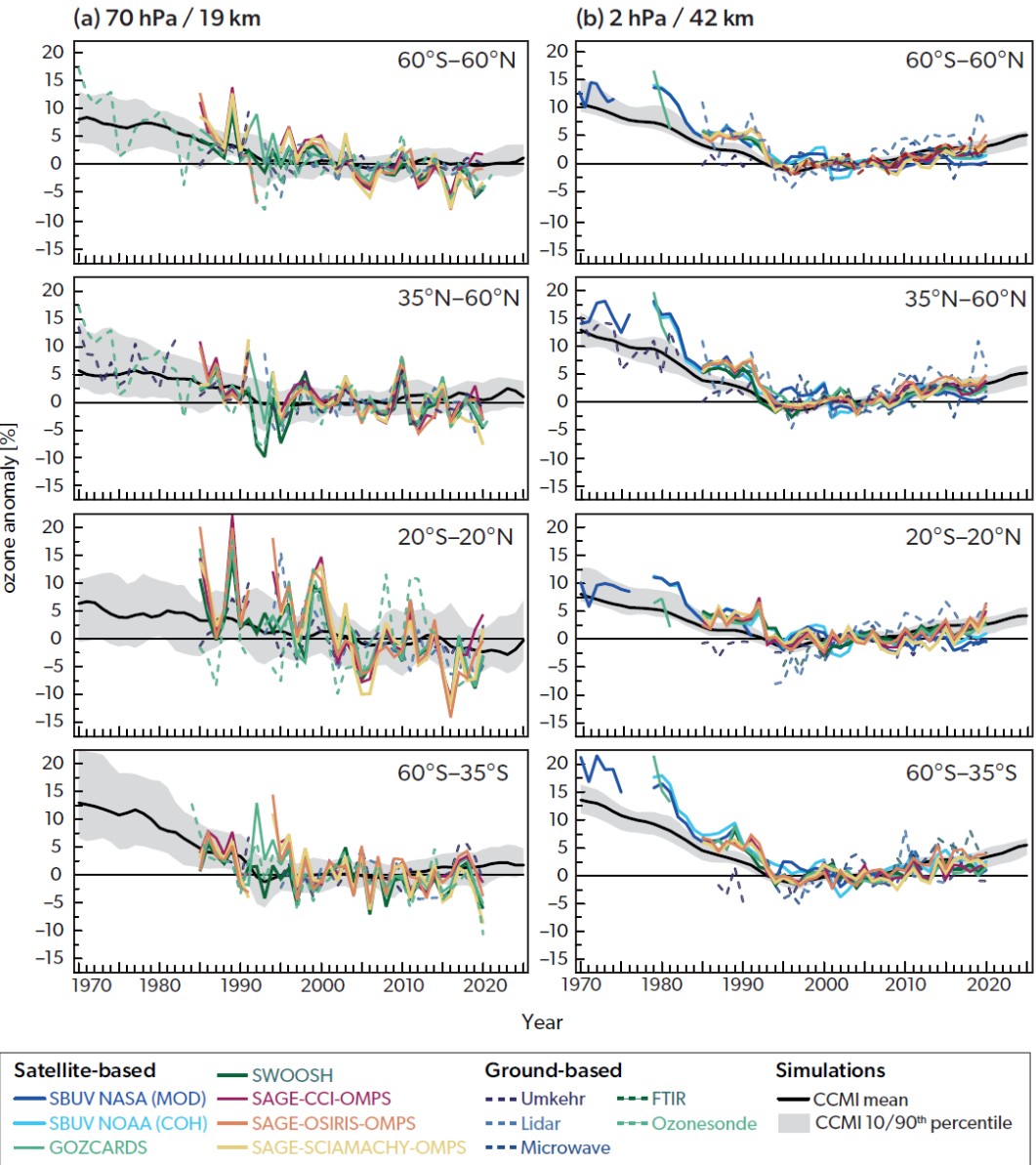

**Figure 2**. Annual mean anomalies of ozone in (a) the lower stratosphere, near 19 km altitude (70 hPa pressure) and (b) the upper stratosphere, near 42 km (2 hPa), for four latitude bands: 60°S–60°N, 35–60°N, 20°S–20°N (tropics), and 60–35°S. Anomalies are referenced to a 1998–2008 baseline. Coloured lines are long-term records obtained by merging data from different nadir (SBUV NASA (MOD) and SBUV NOAA (COH)) or limb-viewing (GOZCARDS, SWOOSH, SAGE-CCI-OMPS, SAGE-OSIRIS-OMPS, SAGE-SCIAMACHY-OMPS) satellite instruments. Dashed coloured lines are long-term records from ground-based observations (Umkehr, lidar, microwave, FTIR and ozonesondes); see Steinbrecht et al. (2017), WMO (2018), and Arosio et al. (2018) for details on the various datasets. The gray shaded areas show the range (10th and 90th percentiles) of 16 CCM simulations performed as part of the CCMI-1 REF-C2 experiment (see Morgenstern et al., 2017) with the black line indicating the median. Taken without change from Figure 3-9 in WMO (2022).

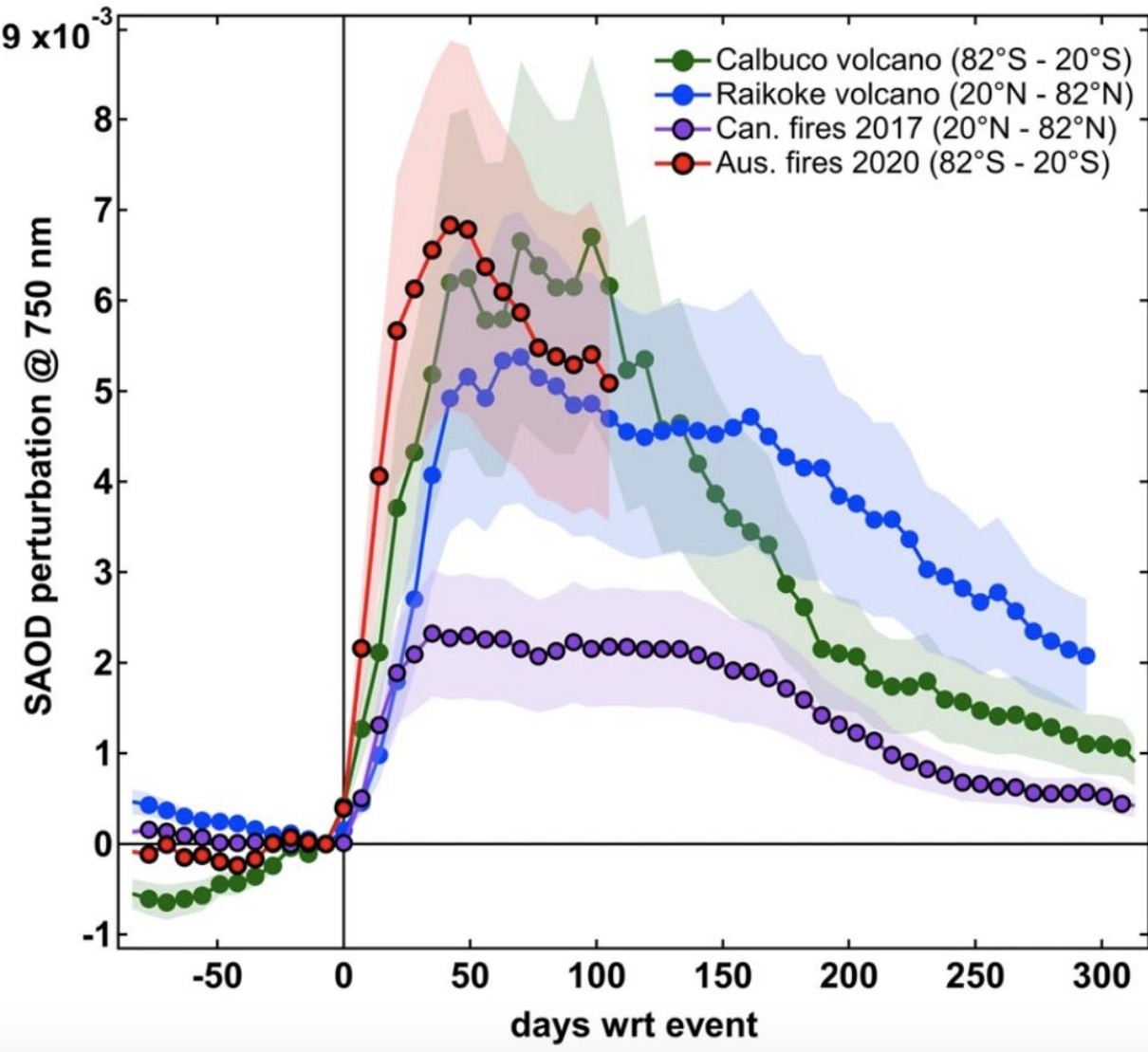

**Figure 3.** Perturbation of the stratospheric aerosol optical depth (SAOD) due to Australian fires and the strongest events since 1991. The curves represent the SAOD perturbation at 746 nm following the Australian wildfires, the previous record-breaking Canadian wildfires in 2017, and the strongest volcanic eruptions in the last 29 years (eruptions of Calbuco volcano in 2015 and Raikoke volcano in 2019). The time series are computed from OMPS-LP aerosol extinction profiles as weekly-mean departures of aerosol optical depth above 380 K isentropic level from the levels on the week preceding the ANY event. The weekly averages are computed over equivalent-area latitude bands roughly corresponding to the meridional extent of stratospheric aerosol perturbation for each event. The shading indicates a 30% uncertainty in the calculated SAOD, as estimated from SAGE III coincident comparisons. Taken from Figure 3 of Khaykin et al. (2020) without change and reproduced under terms of the Creative Commons Attribution 4.0 International Licence (http://creativecommons.org/licenses/by/4.0/).

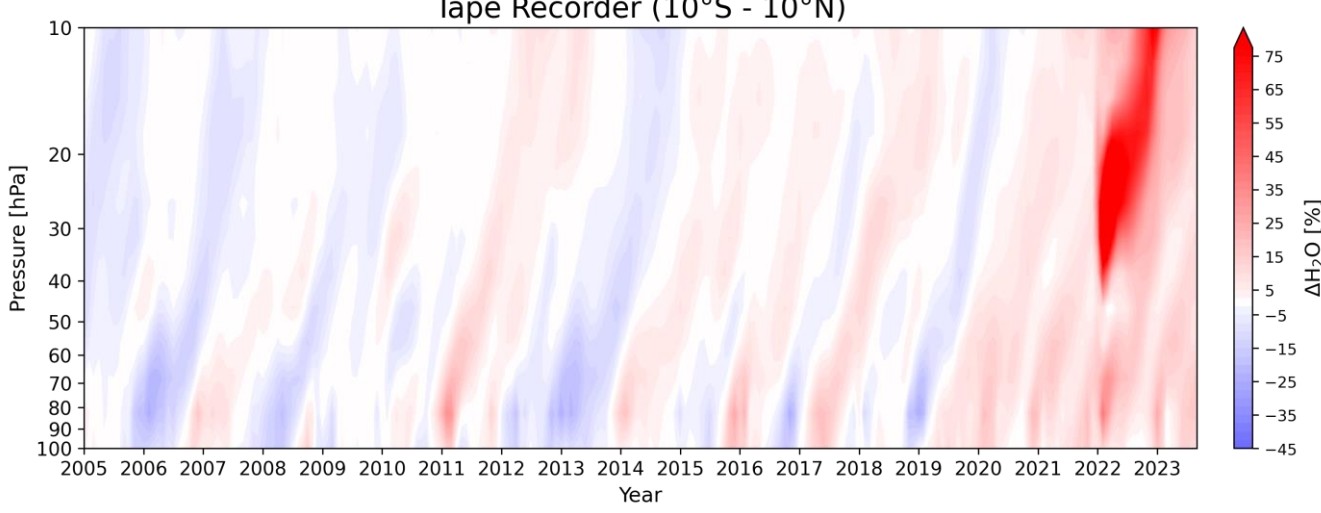

**Figure 4.** Zonal mean $H_2O$ anomalies (%) in the tropics, between 10°S and 10°N (the so-called atmospheric tape recorder) for January 2005 to August 2023. $H_2O$ abundances are based on version 5 Microwave Limb Sounder data. Based on Figure 5a in Millan et al. (2022). Figure courtesy of Xin Zhou (University of Chengdu).