# Peer review of "Opinion: Stratospheric Ozone - Depletion, Recovery and New"

_EGUsphere, 2023_

## Referee Comment (RC1)

Review of EGUsphere 2023-1409

**Opinion: Stratospheric Ozone – Depletion, Recovery and New Challenges**

by Martyn Chipperfield and Slimane Bekki

When I was offered this manuscript for review I gladly agreed, since the title and the authors made me expect a promising contribution. Unfortunately, my expectation was in parts not met. The manuscript is not mature in its present form. It is not balanced and the core of the work does not really contain new insights or ideas for future ozone research. The principle idea for this work is good, but for publication this "opinion" needs to be consolidated and sharpened. The aspects presented and discussed by the authors are so far correct, however the discussed points (i.e. the challenges) must be presented in context: For example, which open scientific questions (regarding the future evolution of the ozone layer) have been on the agenda for years, which are new (recent events), and why they are important for a better understanding of factors influencing the stratospheric ozone layer.

From my point of view, it needs a clear revision before this "opinion" can be published. The overall motivation for writing this opinion is clear, but, as already said, I think that the currently available manuscript is not finished. It needs a clearer structure (outline) and message. In general, I would like to say that this opinion paper, as it has been presented so far, has no real golden thread throughout. At present it is a conglomeration of known information and basic knowledge, but in keeping with the aim of this opinion (i.e. looking at the depletion of stratospheric ozone and the future evolution of the ozone layer in the light of new emerging challenges), a sharpening of the arguments for further (strengthened) observational and modelling efforts is needed. I would expect some (more concrete) ideas for future strategies at the end, for instance which (global) observations are of elementary importance (for monitoring the evolution of the ozone layer) and how Chemistry-Climate Models (CCMs) or Earth-System Models (ESMs) can be used here with an appropriate strategy.

Below are my general points and major caveats.

Section 1, the introduction is very brief. So far, some key references supporting the given statements are missed (for instance at the end of lines 25, 26, 34, 36, 39, 43). I think a short summary of the last 100 years of stratospheric ozone research is very appropriate, including the important contributions of the Nobel Prize winners (Crutzen, Molina and Rowland). From my point of view, the Section 2 (A century of ozone layer research) should be shortened and included in the Introduction part. For instance, the larger passages about Paul Crutzen's work are far too long compared to other parts here (e.g. about the ozone research of Molina and Rowland). There are the (already mentioned) two essays appreciating Crutzen's scientific contributions by Solomon (2021) and Müller (2022), who have already done this excellently. The appropriate references are sufficient. On the other hand, the important contribution by Marcel Nicolet (about the role of HOx with respect to stratospheric ozone in the 1950s) is missing. A short paragraph about the importance/success of the Montreal Protocol including the expected recovery of the ozone layer would be (from my point of view) also fitting in the introduction section.

Then a specific section about current (persistent) uncertainties with respect to stratospheric ozone recovery (over time, regional differences) could focus for instance on the role of climate change (uncertainties of different climate scenarios), the role of VSLS, CCl4, iodine, etc. Another important aspect is the meaning of unforeseen (unexpected) emissions of regulated substances in the Montreal Protocol (the story of CFC11; Montzka et al.), indicating the importance of monitoring ODSs. And, of course, the role of explosive volcanic eruptions in the past (Agung, El Chicon, Pinatubo), which strongly affected stratospheric ozone. The volcanic eruptions of Calbuco and Raikoke must also be discussed here accordingly. Such information is provided (in parts) in Section 3 and the beginning of Section 4.

Furthermore, in a following section, the "newly emerging challenges", i.e. the Australian wildfires (ANY), and the extra-ordinary eruption of Hunga Tongo – Hunga Ha'apai (HTHH) should be discussed in more detail, explaining the scientific (new) challenges, why they need to be addressed and scientifically explored more in depth and that this is also important with regard to basic understanding. The information is (so far) given in Sub-sections 4.2 and 4.3 (a Sub-section 4.1 is missing). But the text passages (paragraphs) are sometimes kept very short, they sometimes seem like individual fragments, unlike a coherent text. The connections need to be better explained.

Finally, these changes would lead to a chapter/section where future activities (incl. measurements, observational capacities, techniques, methods, models) would be suggested and discussed. This part of the paper would be (in my view) the central part of this opinion paper. An opinion about the role of CCMs/ESMs in connection with global observations (monitoring of specific chemical, physical and dynamic quantities) would be helpful, for instance regarding the questions whether such model systems should be prepared in advance for considering "all extra-ordinary" situations, or if it is sufficient that the models can explain the observed features afterwards (a nice example was the explanation of the millennium water drop in the lower tropical stratosphere in 2000 in the following years, e.g. by Randel et al. and other related papers).

Many of the mentioned points in the manuscript are important and correct, but some of them have been thrown together or mixed up. As said in the beginning, it needs a clearer structure and, at the end, a clearer message. This message should be (in my view): Global monitoring of the Earth's atmosphere (i.e. of key-species and other quantities) is absolutely vital and necessary. Numerical models (like CCMs or ESMs) can support the analyses of relevant processes and can help with the interpretation of observations and reveal weaknesses in understanding of the atmospheric system.

---

## Author Response (AR1)

**Response to Reviewers' Comments of EGUsphere 2023-1409**

Opinion: Stratospheric Ozone – Depletion, Recovery and New Challenges

by Martyn Chipperfield and Slimane Bekki

We thank the reviewers for their very helpful comments. These are reproduced below in *italics* followed by our responses (>>). We have improved the manuscript in line with the comments. Some key general points relevant to the comments of both reviewers are:

- We have edited the text to increase (or make clearer) our personal suggestions/opinions on the various topics. Hopefully there is now a clearer thread through the manuscript, starting with the abstract (which was previously just a summary of the topics the article covered, although there is a 250-word limit) through to the revised 'Future Outlook' section.
- In the first draft we aimed to give a much broader summary on the new topics of ANY wildfires and HTHH. This resulted in the somewhat longer Section 4 which we have retained and added detail to.
- As one aspect of the ACP Special Issue is to commemorate the work of Paul Crutzen, we were specifically asked to summarise his work on stratospheric ozone. We note this in the Introduction. Hence we still have Section 2 on past research with a large focus on Crutzen's stratospheric work.

**Reviewer 1**

*When I was offered this manuscript for review I gladly agreed, since the title and the authors made me expect a promising contribution. Unfortunately, my expectation was in parts not met. The manuscript is not mature in its present form. It is not balanced and the core of the work does not really contain new insights or ideas for future ozone research. The principle idea for this work is good, but for publication this "opinion" needs to be consolidated and sharpened. The aspects presented and discussed by the authors are so far correct, however the discussed points (i.e. the challenges) must be presented in context: For example, which open scientific questions (regarding the future evolution of the ozone layer) have been on the agenda for years, which are new (recent events), and why they are important for a better understanding of factors influencing the stratospheric ozone layer.*

>> We now refer to 'well established' and 'new' challenges. We have tried to emphasise the importance of these.

*From my point of view, it needs a clear revision before this "opinion" can be published. The overall motivation for writing this opinion is clear, but, as already said, I think that the currently available manuscript is not finished. It needs a clearer structure (outline) and message. In general, I would like to say that this opinion paper, as it has been presented so far, has no real golden thread throughout. At present it is a conglomeration of known*

*information and basic knowledge, but in keeping with the aim of this opinion (i.e. looking at the depletion of stratospheric ozone and the future evolution of the ozone layer in the light of new emerging challenges), a sharpening of the arguments for further (strengthened) observational and modelling efforts is needed. I would expect some (more concrete) ideas for future strategies at the end, for instance which (global) observations are of elementary importance (for monitoring the evolution of the ozone layer) and how Chemistry-Climate Models (CCMs) or Earth-System Models (ESMs) can be used here with an appropriate strategy.*

>> We have now tried to make our opinions clearer throughout and draw them together in a bulleted list of 4 points in the final section.

*Below are my general points and major caveats.*

*Section 1, the introduction is very brief. So far, some key references supporting the given statements are missed (for instance at the end of lines 25, 26, 34, 36, 39, 43). I think a short summary of the last 100 years of stratospheric ozone research is very appropriate, including the important contributions of the Nobel Prize winners (Crutzen, Molina and Rowland). From my point of view, the Section 2 (A century of ozone layer research) should be shortened and included in the Introduction part. For instance, the larger passages about Paul Crutzen's work are far too long compared to other parts here (e.g. about the ozone research of Molina and Rowland). There are the (already mentioned) two essays appreciating Crutzen's scientific contributions by Solomon (2021) and Müller (2022), who have already done this excellently. The appropriate references are sufficient. On the other hand, the important contribution by Marcel Nicolet (about the role of HOx with respect to stratospheric ozone in the 1950s) is missing. A short paragraph about the importance/success of the Montreal Protocol including the expected recovery of the ozone layer would be (from my point of view) also fitting in the introduction section.*

>> We have added in more background references. We realise that the work of Crutzen is discussed to a much greater extent than other researchers but that was in the remit we were given by ACP (see above) and so we have kept a separate Section 2. We have added in the reference of Nicolet (1970) for HOx.

We already mentioned the Montreal Protocol in the Introduction. We have added a sentence about is importance/success: "Nevertheless, the Protocol can therefore be considered on track in its aim of protecting the ozone layer from the effects of halogenated ODSs".

*Then a specific section about current (persistent) uncertainties with respect to stratospheric ozone recovery (over time, regional differences) could focus for instance on the role of climate change (uncertainties of different climate scenarios), the role of VSLS, CCl4, iodine, etc. Another important aspect is the meaning of unforeseen (unexpected) emissions of regulated substances in the Montreal Protocol (the story of CFC11; Montzka et al.), indicating the importance of monitoring ODSs. And, of course, the role of explosive volcanic eruptions in the past (Agung, El Chicon, Pinatubo), which strongly affected stratospheric*

*ozone. The volcanic eruptions of Calbuco and Raikoke must also be discussed here accordingly. Such information is provided (in parts) in Section 3 and the beginning of Section 4.*

>> The information described by the reviewer is contained in Sections 3 and 4. We feel the split between halogenated gases relevant to the MP (Section 3) and other factors (Section 4) is one logical way to split this information. Hence we have kept this structure rather than the alternative 'persistent/new' split suggested by the reviewer. However, we do now use the terms 'well established' and 'new'.

We have increased the discussion of climate change in Section 4, in particular to reflect the uncertainty in GHG scenarios, as distinct from lack of understanding of atmospheric processes. We have also added more details into the discussion of ANY and HTHH.

*Furthermore, in a following section, the "newly emerging challenges", i.e. the Australian wildfires (ANY), and the extra-ordinary eruption of Hunga Tongo – Hunga Ha'apai (HTHH) should be discussed in more detail, explaining the scientific (new) challenges, why they need to be addressed and scientifically explored more in depth and that this is also important with regard to basic understanding. The information is (so far) given in Sub-sections 4.2 and 4.3 (a Sub-section 4.1 is missing). But the text passages (paragraphs) are sometimes kept very short, they sometimes seem like individual fragments, unlike a coherent text. The connections need to be better explained.*

>> We have added some more details and tried to make the text more coherent.

*Finally, these changes would lead to a chapter/section where future activities (incl. measurements, observational capacities, techniques, methods, models) would be suggested and discussed. This part of the paper would be (in my view) the central part of this opinion paper. An opinion about the role of CCMs/ESMs in connection with global observations (monitoring of specific chemical, physical and dynamic quantities) would be helpful, for instance regarding the questions whether such model systems should be prepared in advance for considering "all extra-ordinary" situations, or if it is sufficient that the models can explain the observed features afterwards (a nice example was the explanation of the millennium water drop in the lower tropical stratosphere in 2000 in the following years, e.g. by Randel et al. and other related papers).*

>> We have renamed the final section as 'Future Outlook'. It tries to draw the points together for a concise summary of suggested future work (though some of the points mentioned by the reviewer are made in previous sections).

The reviewer raises an interesting point about the role of ESMs. These are complex and very costly models. If all components are included then the number of simulations that can be performed would be very limited. We would argue that it is better to have substantial number of model simulations (ensemble members, different scenarios) of a model with the relevant stratospheric processes in order to get the most robust estimates for the ozone layer. We discuss this in Section 6. Therefore, in relation to the question above we do not think that models can be prepared in advance to cover all possible situations. Processes may

not be known about, or may be unimportant unless the atmosphere is subject to some extreme and unexpected forcing.

*Many of the mentioned points in the manuscript are important and correct, but some of them have been thrown together or mixed up. As said in the beginning, it needs a clearer structure and, at the end, a clearer message. This message should be (in my view): Global monitoring of the Earth's atmosphere (i.e. of key-species and other quantities) is absolutely vital and necessary. Numerical models (like CCMs or ESMs) can support the analyses of relevant processes and can help with the interpretation of observations and reveal weaknesses in understanding of the atmospheric system.*

>> OK. We have tried to tidy up the structure and flow of the paper and to have more continuity for the opinions. The final 'Future Outlook' section now has the 4 bullet points on research challenges. We have added stronger 'opinion' statements to the abstract (subject to word limit) to set the scene from the start. Thank you for the suggested message. We have used an adaptation of this at the end of the paper.

**Reviewer 2**

*This opinion piece on stratospheric ozone is an interesting read. The authors briefly recount the basics of the ozone story, the Montreal Protocol and ozone recovery along with the substantial contributions of Paul Crutzen. They further discuss more recent events like the Australian wildfire, the Hunga Tonga eruption, observations from space and modelling challenges. The authors are well qualified to discuss these topics. As such, the manuscript will be acceptable for publication after the authors consider the comments below.*

>> Thank you for the summary.

1) *The text is uneven as an opinion piece. Section 3, 5 and 6 have a number of valuable opinion statements (e.g., ln122, ln147, ln157, ln393, ln411, ln421, ln454). In contrast, they are lacking in Section 4 which reads more as a tutorial of recent impacts on the stratosphere. I recommend adding opinions to Sect 4 or shortening considerably and relying on references to describe these processes.*

>> To be consistent with the comments of Reviewer 1, who requests more detail in Section 4, we have added opinions here and tried to make clearer the reason for the longer section.

*And the impact of the manuscript would be enhanced if the authors review the text to make sure their existing opinions clearly stated and consider where opinions could be added and that the combined portfolio of opinions has a reasonable coherence and story line.*

>> Thank you for this major comment. As noted above, we now try to have a clear thread through the manuscript with opinions in the overarching topics of observations and modelling.

2) *The conclusions 'threaten this essential component' and 'threaten further depletion' seem overstated since they are not usefully quantified or justified. I recommend eliminating that word without further explanation.*

>> OK. We have changed to 'can perturb'.

3) *The role of greenhouse gas (GHG) increases in influencing future ozone is not well discussed, especially since large GHG increases will lead to ozone super recovery for which the impacts are poorly known. Suggest emphasizing that ODS and ODS substitute emissions also enhance climate change which will influence ozone in multiple ways.*

>> We have added more on climate change to Section 4.

4) *A useful additional opinion would be to note the importance of communicating ODS and ozone layer science to policy makers to guide future decision making to protect ozone and climate. For example, the 'effectiveness' comment on ln 142 implicitly includes the effective communication of scientists and policymakers over a number of years after the emissions were documented.*

>>OK, thank you. We have added a sentence.

5) *The statement "However, recent discoveries related to increased emissions of controlled ODSs and uncontrolled shorter-lived halogenated source gases have raised some concerns on the continued success of the treaty and the outlook for ozone recovery." seems alarmist. A more valid perspective, for example, is that the success of dealing with CFC-11 emissions has uniquely demonstrated the resilience of the Mont Prot, the effectiveness of their provisions, and the importance of continued vigilance of atmospheric abundances.*

>> We have added references to this section and the sentence 'However, the success of dealing with the CFC-11 issue (see below) has demonstrated the resilience of the protocol, the effectiveness of its provisions, and the importance of continued vigilance regarding atmospheric trace gases.'

6) *As an aside, a real vulnerability of the Mont Prot is that the atmospheric observations of ODSs and other gases that provide essential information for the foundation of MP regulations are not controlled/supported/managed by the Mont Prot. Instead, they are provided independently by governments, ie NOAA and AGAGE networks.*

*Suggest pointing to the white paper prepared in part by the Scientific Assessment Panel of the Mont Prot that offers recommendations to mitigate critical gaps in ODS observations. The white paper was motivated by the discovery of unreported CFC-11 emissions by the global networks. https://ozone.unep.org/system/files/documents/ORM11-II-4E.pdf*

>> OK. The reviewer is offering this comment as an 'aside'. Yes, we agree that it is essential to have effective observations of the controlled ODSs and other potentially important gases. That is something that we try to emphasise in our opinions. However, we do not feel that our manuscript is the place to go into details about how these observations are provided.

We have added the reference at the end of Section 5 (ORM, 2021b) using the web link for the full ORM meeting documents.

6a) *Section 5 has an extremely important and timely message about continued observations that could be elevated to the abstract. It is truly unfortunate that, for example, the HT impacts diagnosed in the Santee et al paper cited below will not be repeatable for a future eruption after the MLS instrument is retired.*

>> Yes. The new abstract has the sentence 'We will, in effect, be largely blind to the stratospheric effects of large wildfires or volcanic eruptions in the near future'.

**Other comments:**

7) *Suggest adding a reference to the Fishman et al tribute to Crutzen in BAMS (2022).*

>>OK. We have added a reference to Fishman et al. (2023).

8) *ln14 change "Further unexpected perturbations to the ozone layer are occurring at the moment through injection of very large amounts…'' to "Further perturbations to the ozone layer are occurring at the moment through unexpected injection of very large amounts…''*

>>OK. We have moved the word 'unexpected'.

9) *ln116: Suggest changing to: "ODS production controls have caused a net reduction in the tropospheric source gases (Figure 1a) that transport chlorine and bromine to the stratosphere."*

>> OK. Changed to 'Controls on ODS production have caused a net reduction in the tropospheric source gases (Figure 1a) that…'

10) *ln125 Suggest adding clarification that dichloromethane has a substantial non-anthropogenic source.*

>> We think that this is incorrect. Maybe a typo and the reviewer was suggesting that we say DCM is mainly anthropogenic. We have added 'which is mainly of anthropogenic origin'.

11) *ln132 Suggest changing to "The history of the MP since its signing in 1987 (and ratification in 1989) is one of continued success – as evidenced…"*

>> OK. Text has been changed.

12) *ln144 change to 'ODSs' and delete 'it'*

>> OK. Both changes made.

13) *ln163 Change 'can be' to 'was'*

>> We were making the general point of how this is done, not a specific reference to a method used in a paper. We have changed 'can be' to 'is'.

14) *ln 189 Recovery estimates already include ongoing and future ODS abundances and emissions. Suggest changing to 'Clearly, ongoing emissions of chlorine and bromine from ODSs or VSLS that are not already accounted for will act to slow down the estimated rate of recovery…'*

>> OK, thank you for this correction. We have added 'that are not already accounted for'.

15) *ln235 Suggest changing to '…relatively limited impact on global ozone and, unlike the anthropogenic halogen emissions, are only expected to…'*

>> OK. Changed 'threat' to 'emissions'.

16) *ln249 Suggest 'when aerosols are enhanced'. Suggest choosing a more contemporary reference than Hofmann and Solomon.*

>> We think that this is a suitable reference because it was the first to explain with observations and model simulations the impact of Pinatubo aerosol heterogeneous chemistry on the stratosphere. The reference can be easily found and we have several which are older than this.

17*) ln262 Suggest citing how recent observed stratospheric aerosol perturbations have offset climate change with the recent Yu et al. paper, which derives in part from Fig. 3 of this manuscript: Yu, P., et al. (2023). Radiative forcing from the 2014–2022 volcanic and wildfire injections. Geophysical Research Letters, 50, e2023GL103791.*
*https://doi.org/10.1029/2023GL103791*

>> OK, reference added.

18) *ln332 Suggest citing the recent Santee et al paper which diagnoses the HT impact on het processes that alter ozone chemistry. Santee, M. L., et al. (2023). Strong evidence of heterogeneous processing on stratospheric sulfate aerosol in the extrapolar Southern Hemisphere following the 2022 Hunga Tonga-Hunga Ha'apai eruption. Journal of Geophysical Research: Atmospheres, 128, e2023JD039169. https://doi. org/10.1029/2023JD039169*

>> OK. We have added this reference to Section 4.2.2.

19) *ln392 Suggest changing to 'In recent years we have benefitted from a wealth of observations from instruments in ground-based networks and on balloon, aircraft and satellite platforms.'*

>>OK. Text has been edited as suggested.

**20)** *It would be great to update the H2O time series in Fig. 4 as a private communication*

>> We have created our own update to this figure just based on MLS data which spans from 2005 onwards and is updated to August 2023. This is sufficient to show the large perturbation from Hunga Tonga.

**20b)** *ln412 suggest replacing 'that we have been used to' to 'that has been so valuable' or 'that has been so essential to understanding ozone depletion' or similar.*

>> OK We have changed this to 'that has been so essential to understanding the ozone layer…'.

**21)** *ln426 Suggest omitting 'and simulators' since it is a tool.*

>> OK. Words deleted.

**22)** *ln814. This figure is outdated. Suggest updating to Fig ES-1 of the 2022 assessment.*

>> We disagree that this figure is outdated for our purposes. The 2018 figure is based on results from ~20 CCMs performing specific simulations to model recovery of the ozone layer. The 2022 figure is based on 6 CMIP6 models (i.e. much fewer models and without a specific focus on the stratosphere). Also, the 2022 figure shows a dependence of Antarctic recovery on GHG loading that was not apparent in earlier studies. As discussed in WMO (2022) this may be due to the small number of models or details of the specific GHG scenarios used in the SSPs. One thing that we want to make clear(er) in the revised paper is the need for robust, well-tested CCMs to be used for these studies. For various reasons, that was probably not the case in WMO (2022). Thus, we have kept our '2018' figure in order to back up our messages.

**Community Comment #1 by Albert Ansmann.**

We thank you for your comment on the importance of wildfire smoke and for alerting us to your work related to wildfire observations and ozone loss at high northern and southern latitudes. Specifically, you referenced the following three papers:

- Ansmann, A., Ohneiser, K., Chudnovsky, A., Knopf, D. A., Eloranta, E. W., Villanueva, D., Seifert, P., Radenz, M., Barja, B., Zamorano, F., Jimenez, C., Engelmann, R., Baars, H., Griesche, H., Hofer, J., Althausen, D., and Wandinger, U.: Ozone depletion in the Arctic and Antarctic stratosphere induced by wildfire smoke, Atmos. Chem. Phys., 22, 11701–11726, https://doi.org/10.5194/acp-22-11701-2022, 2022.
- Ohneiser, K., Ansmann, A., Chudnovsky, A., Engelmann, R., Ritter, C., Veselovskii, I., Baars, H., Gebauer, H., Griesche, H., Radenz, M., Hofer, J., Althausen, D., Dahlke, S., and Maturilli, M.: The unexpected smoke layer in the High Arctic winter stratosphere during MOSAiC 2019–2020 , Atmos. Chem. Phys., 21, 15783–15808, https://doi.org/10.5194/acp-21-15783-2021, 2021.
- Ohneiser, K., Ansmann, A., Kaifler, B., Chudnovsky, A., Barja, B., Knopf, D. A., Kaifler, N., Baars, H., Seifert, P., Villanueva, D., Jimenez, C., Radenz, M., Engelmann, R., Veselovskii, I., and Zamorano, F.: Australian wildfire smoke in the stratosphere: the decay phase in 2020/2021 and impact on ozone depletion, Atmos. Chem. Phys., 22, 7417–7442, https://doi.org/10.5194/acp-22-7417-2022, 2022

As explained in our response to the two anonymous reviewers, we have expanded our discussion of wildfire smoke. Your 2022 papers are the ones most relevant to our discussion on ANY and we have added citations to them

**Response to Community Comments of EGUsphere 2023-1409**

Opinion: Stratospheric Ozone – Depletion, Recovery and New Challenges

by Martyn Chipperfield and Slimane Bekki

Community Comment #1 by Albert Ansmann.

We thank you for your comment on the importance of wildfire smoke and for alerting us to your work related to wildfire observations and ozone loss at high northern and southern latitudes. Specifically, you referenced the following three papers:

- Ansmann, A., Ohneiser, K., Chudnovsky, A., Knopf, D. A., Eloranta, E. W., Villanueva, D., Seifert, P., Radenz, M., Barja, B., Zamorano, F., Jimenez, C., Engelmann, R., Baars, H., Griesche, H., Hofer, J., Althausen, D., and Wandinger, U.: Ozone depletion in the Arctic and Antarctic stratosphere induced by wildfire smoke, Atmos. Chem. Phys., 22, 11701–11726, https://doi.org/10.5194/acp-22-11701-2022, 2022.
- Ohneiser, K., Ansmann, A., Chudnovsky, A., Engelmann, R., Ritter, C., Veselovskii, I., Baars, H., Gebauer, H., Griesche, H., Radenz, M., Hofer, J., Althausen, D., Dahlke, S., and Maturilli, M.: The unexpected smoke layer in the High Arctic winter stratosphere during MOSAiC 2019–2020 , Atmos. Chem. Phys., 21, 15783–15808, https://doi.org/10.5194/acp-21-15783-2021, 2021.
- Ohneiser, K., Ansmann, A., Kaifler, B., Chudnovsky, A., Barja, B., Knopf, D. A., Kaifler, N., Baars, H., Seifert, P., Villanueva, D., Jimenez, C., Radenz, M., Engelmann, R., Veselovskii, I., and Zamorano, F.: Australian wildfire smoke in the stratosphere: the decay phase in 2020/2021 and impact on ozone depletion, Atmos. Chem. Phys., 22, 7417–7442, https://doi.org/10.5194/acp-22-7417-2022, 2022

As explained in our response to the two anonymous reviewers, we have expanded our discussion of wildfire smoke. Your 2022 papers are the ones most relevant to our discussion on ANY and we have added citations to them